# Health-related quality of life and its associated factors among patients with asthma: A multi-centered cross-sectional study in selected referral hospitals in Northwest Ethiopia

**Eyayaw Ashete Belachew**[1]*, **Ashenafi Kibret Sendekie**[1], **Sumeya Tadess**[1], **Mekuriaw Alemayehu**[2]

**1** Department of Clinical Pharmacy, School of Pharmacy, College of Medicine and Health Sciences, University of Gondar, Gondar, Ethiopia, **2** Institute of Public Health, College of Medicine and Health Sciences, University of Gondar, Gondar, Ethiopia

* eyayawashete2016@gmail.com

**Data Availability Statement:** All relevant data are within the paper and its Supporting Information files.

## Abstract

### Background

Patients with asthma have a compromised health-related quality of life (HRQoL) due to factors related sociodemographic, clinical, and environmental factors. This study assessed the HRQoL and its determinants in patients with asthma in selected public referral hospitals in Northwest Ethiopia.

### Design, setting, and participants

A multicenter facility-based cross-sectional study was conducted in selected hospitals in Northwest Ethiopia from August to October 2021. Participants were enrolled in the study using a systematic random sampling technique.

### Main outcome measures

HRQoL was assessed using the asthma-specific quality of life tool Mini-Asthma Quality of Life Questionnaire (Mini-AQLQ). Simple and multivariable linear regression analyses were conducted to determine the association between independent variables and HRQoL. A p-value of < 0.05 at 95% CI was considered statistically significant.

### Results

A total of 409 patients were included in the final analysis, and more than half (59.2%) of the subjects had a good health-related quality of life. Regarding HRQoL determinants, asthma control score ($\beta$ = 0.14, 95% CI: 0.09, 0.17; p 0.001), insurance user ($\beta$ = 0.15, 95% CI: 0.01, 0.29); p = 0.042), the high role of patient enablement ($\beta$ = 0.39, 95% CI: 0.25, 0.54; p 0.001), belief in asthma medication ($\beta$ = -0.23, 95% CI: -0.36, -0.10;p = 0.001), non-adherence to guidelines ($\beta$ = -0.30, 95% CI: -0.47, -0.15; p < 0.001), and being homemaker ($\beta$ = -0.21, 95% CI: -0.39, -0.01; p = 0.040) were the significant predictors of HRQoL.

**Funding:** The authors received no specific funding for this work.

**Competing interests:** The authors declared That they have no Competing interest.

**Abbreviations:** ACT, Asthma Control Test; AQLQ, Asthma Quality of Life Questionnaire; BMQ, Beliefs about Medicine Questionnaire; CCI, Charlson Comorbidity Index; COPD, Chronic Obstructive Pulmonary Disease; GINA, Global Initiative for Asthma; HRQoL, Health Related Quality of Life; ICS, Inhaled Corticosteroids; LABA, Long-Acting Beta-2 Agonists; mPEI, Modified Patient Enablement Index; MARS-A, Medication Adherence Rating Scale; QOL, Quality of Life; SABA, Short Acting Beta-2 Agonists; WHO, World Health Organization.

## Conclusion

Overall, more than half of the study participants were found to have good HRQoL. HRQoL among adults with asthma was largely dependent on the level of asthma control. Socio-demographic, clinical, healthcare-related, and medication-related variables were significantly associated with health-related quality of life. Therefore, healthcare providers should include comprehensive asthma education along with an integrated treatment plan to improve asthma control and the HRQoL of patients.

## Introduction

Asthma is a chronic inflammatory airway disease characterized by a heterogeneous illness; symptoms include wheezing, shortness of breath, tightness of the chest, coughing, and restriction of expiratory airflow, affecting daytime and nighttime activities [1]. Environmental pollution, an upper respiratory infection, bugs in the home, a cold, laughter, cigarette smoke, or a strong odor could exacerbate it [2].

Globally, asthma is a big healthcare concern and the 14th most important disorder in terms of the extent and duration of disability [3]. According to a 2018 report, it is estimated to affect more than 339 million people globally [4]. In Africa, the prevalence in the total population increased from 74.4 million to 119.3 million within just two decades (1990–2010) [5]. Reports from sub-Saharan African countries also showed a surge in prevalence [5, 6].

Health-related quality of life (HRQoL) is a multidimensional concept that includes global health perspectives, symptom status, functional status, biological and physical variables, individual and environmental characteristics, and general health perception [7]. According to the World Health Organization (WHO), "quality of life" (QOL) is defined as an individual's perception of their position in life in the context of the culture and value systems in which they live and in relation to their goals, expectations, standards, and concerns [8]. This definition considers individuals' satisfaction with the physical, psychological, social, environmental, and spiritual aspects of their lives [9]. The quality of life is the overall sense of well-being, including aspects of happiness and satisfaction with life [10]. It is a broader term and remains subjective [11].

Several clinical, sociodemographic, treatment-related, and behavioral factors influence asthma patients' health-related quality of life. Female gender, level of asthma control, co-morbidities, healthcare provider adherence to guideline usage, and use of healthcare insurance are some factors that predict HRQoL in patients with asthma [7, 12–18].

Several studies have used generic quality of life instruments to assess the burden of asthma as perceived by the patient, but there are now several asthma-specific quality of life measurement tools that include items measuring the individual's functional status (ability to perform daily functions; limitations on daily or desired activities) or health status (frequency and intensity of asthma symptoms; need for a short-acting -agonist (SABA)) [19]. Many studies have provided valuable information on the HRQoL of patients with asthma around the worldwide [7, 14, 15, 20–25]. However, there is a knowledge gap in the study area regarding the HRQoL of asthmatic patients. As a result, the current study examined the level of HRQoL and its predictors in adults with asthma in northwestern Ethiopia.

## Methods and materials

### Study setting and design

A multi-centered, institutional-based, cross-sectional survey was conducted at three public comprehensive specialized hospitals from August to October 2021. The selected hospitals were

the University of Gondar Comprehensive Specialized Hospital (UoGCSH), Felege Hiwot Comprehensive Specialized Hospital (FHCSH), and Tibebe-Ghion Comprehensive Specialized Hospital (TGCSH). These hospitals were selected randomly from the region's public and private hospitals.

The UoGCSH is located in Gondar City, approximately 750 km from Addis Ababa, Ethiopia's capital city. According to UOGCSH records, the asthma clinic follows up every on Monday and reviews at least 280–300 asthmatic patients monthly, for 900 asthma patients. The second hospital, FHCSH, is located in Bahir Dar, 565 kilometers northwest of Ethiopia's capital. The hospital's chest clinic treats patients with chronic asthma and chronic obstructive pulmonary disease (COPD). The asthma follow-up runs from Monday to Friday, and FHCSH reviews the medical records of at least 190–200 patients per month on average. TGCSH is a teaching hospital affiliated Bahir Dar University's college of Medicine and health sciences and is located in Bahir Dar, Ethiopia. Asthma follow-up from the outpatient department occurs every on Wednesday, and an average, 70–80 patients per month are reviewed as TGCSH medical records.

### Study population and sampling

This study included patients with asthma aged 18 and older who were receiving follow-up care at the designated hospitals. In addition, to be eligible for the study, participants must have received inhaled corticosteroid (ICS) therapy within the previous three months. However, patients who were unable to communicate during the interview were excluded.

The sample size was determined using a single population proportion formula, considering the following assumptions: $n = p*(1-p) * Z^2/W^2$, where the proportion (p) was set at 50% because there had been no previous study in the study area to measure HRQoL among asthmatic patients, the absolute precision or margin of error was set at w = 0.05 (5%), Z = 1.96 at a 95% confidence level, and a 10% contingency for non-response was used. Thus, the final sample size is 422. The sample size was assigned to each hospital in proportion to the number of patients based on their previous three-month records. The study participants were then proportionally assigned to each hospital. UOGCSH, FHCSH, and TGCSH were performed on 224, 142, and 56 patients, respectively, in this study (**Fig 1**).

Finally, the sampling fraction (k-interval) is 1695/422 = 4 because, the sample was collected within three months. The initial study subject was chosen by lottery, and then each of the four participants approached the study individuals and collected their corresponding medical

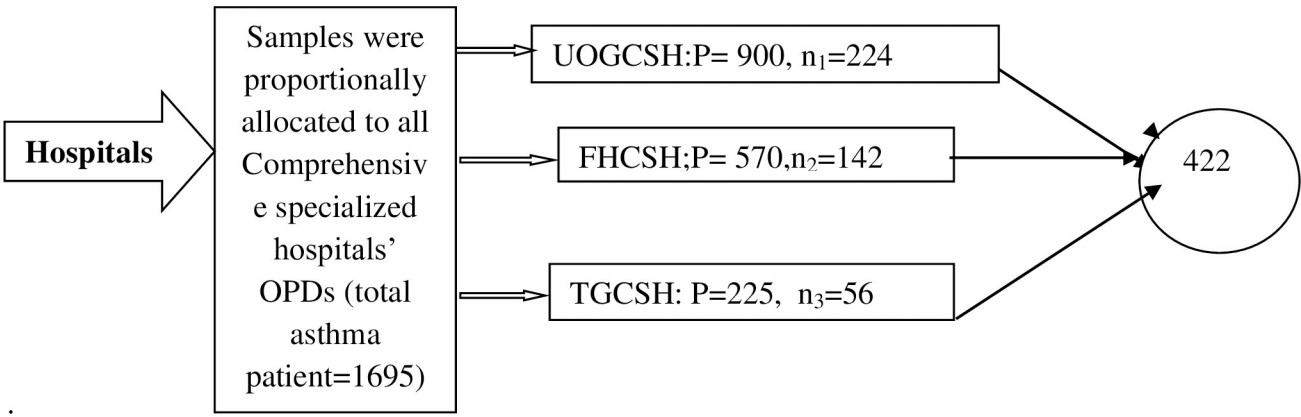

**Fig 1. Proportional allocation of participants in the three hospitals.**

records and relevant data. Furthermore, the chosen respondent was interviewed. The medical records of the study subjects who met the inclusion criteria were reviewed for this study. When one of the medical records on hand was deemed ineligible, the next closest one was chosen, and this method was followed throughout the data collection procedure.

## Data collection tools

The Asthma Control Test (ACT) and the Mini-AQLQ tools were used to assess asthma control and HRQoL, respectively. People from all across the world can use these validated questionnaires [26, 27]. The ACT tool is a simple test used to assess asthma control in patients aged 12 and up. It includes five 5-point scale questions about the frequency of asthma symptoms and the use of rescue medication in the previous four weeks by participants [26]. The overall score ranged from 5 (worst control) to 25 (complete control). The mini-AQLQ is a disease-specific tool for assessing asthma-related HRQoL. It included 15 questions divided into four sections: symptoms, activity limitations, emotional function, and environmental stimuli. Before to enrolling in the study, participants were asked about their health. The researchers were classified with each of the 15 questions on a 7-point Likert scale. The mean of the 15 responses was used to calculate the overall mini-AQLQ score, whereas the mean of responses to items in each domain was used to calculate the domain scores [27]. The Cronbach alpha results for the tools were 0.90 for the mini-AQLQ and 0.83% for the ACT.

The Medication Adherence Rate Scale (MARS-A) was used to assess patient medication adherence [28]. The MARS-A, a self-reported adherence tool that demonstrated good test-retest reliability (r = 0.65, p-value = 0.001), internal consistency reliability of 0.85, sensitivity of 0.82, and specificity of 0.69, was used to assess ICS adherence. The responses were scored on a 5-point Likert scale (1 = never, 2 = rarely, 3 = occasionally, 4 = frequently, and 5 = always). The average score of the 10 items (1–5) is used to calculate self-reported adherence, with higher scores indicating higher levels of reported adherence. A MARS-A score of 4.5 or higher was defined as high self-reported adherence [29]. The Cronbach alpha was performed for the tool, resulting in a value of 0.90.

An older version of the Beliefs about the Medicine Questionnaire (BMQ) was used to assess patient medication beliefs [30]. The Belief about Medicine Questionnaire is a 10-item questionnaire designed to assess patients' beliefs about their prescribed medication. It is made up of two five-item scales: specific necessity and concern. The specific-necessity scale assesses patients' beliefs about the medication prescribed to maintain their health now and in the future. The specific concern scale, on the other hand, assesses patients' perceptions of the negative consequences of taking medications, such as long-term effects and dependence [31]. Patients' beliefs about medicine before and after clinical appointments regarding their asthma management care were calculated using a 5-point Likert-type scale ranging from strongly disagree (score = 1) to strongly agree (score = 5) [32]. Then, the scores from individual items within both scales were summed. As a result, the overall total scores for the necessity and concern scales range from 5 to 25. Finally, higher scores indicated stronger beliefs. The internal validity of the tool was checked, resulting in a Cronbach alpha of 0.73.

The modified patient enablement index (mPEI) was used to assess the patient's enablement. The mPEI is a six-item questionnaire, with each item scored on a Likert scale from 0 (less or not applicable) to 2 (much better). As a result, the score ranged from 0 (the lowest) to 12 (the highest). A higher score reflects higher patient enablement, and a score of more than 6 indicates clinically meaningful enablement. It includes six items that assess the patient's ability to deal with life, understand the illness, cope with it, stay healthy, maintain confidence in one's health, and help oneself [33]. The tool's Cronbach alpha was calculated to be 0.92.

The Charlson Comorbid Index (CCI) was used to calculate the burden of co-morbidity on asthma control and HRQOL [34]. It was divided into mild, moderate, and severe categories based on co-morbidity index scores of 1, 2, 3, and 4, respectively.

Additionally, other studies were reviewed to develop questions [2, 35]. A structured questionnaire was used to collect factors associated with HRQoL, which included socio-demographic factors, modifiable and non-modifiable factors, triggers, patient compliance, and clinical factors that influence asthma control and, as a result, HRQoL.

All the mentioned instruments were prepared in English and then translated into Amharic by English- and Amharic-speaking persons. The translation was verified for compatibility with the original version through a process of forward and backward translation. The face validity of the survey was assessed among three clinical pharmacy teachers for the clarity of the questions. Then, the survey was assessed for content, design, readability, and comprehension by 5% of the study participants, and socio-cultural adoptions were made using WHO recommendations with modifications based on responses, so that the survey was simple to understand and answer yet providing accurate data.

## Data collection procedure

For the study population, the data collectors interviewed the participants and mitigated against such obstacles as language barriers and low literacy levels. The Amharic version of the tool was used. The principal investigator (PI) selected six data collectors (two for each study area) and trained them for two days before the commencement of the research. The data collectors were nurses working in ambulatory care at UOGCSH, FHCSH chest clinic, and TGCSH ambulatory care. The training entailed an explanation of the study, its objectives, and its importance. There was demonstration and practical training on how to use the data collection tools. Ethical considerations and overall expectations from the scientific research were explained. The competence of the data collectors was assured by the PI during pre-testing before the study commenced. This was accomplished by evaluating how accurately the collector extracted data and completed the questionnaires. Where further training was required, it was provided and reinforced until competence was ascertained. Finally, after the data were checked for completeness, it was cleaned and analyzed. A pre-test on the first 5% of the sample size was conducted. Based on the results obtained, the questionnaire was modified.

## Data analysis and management

The data was checked for its completeness and cleanliness, then coded and entered into the Epi-Info Version 7 database and exported to SPSS Version 26 for analysis. Descriptive statistics such as the mean, median, and proportion, were used to describe the characteristics of the study participants and were displayed as tables and figures. Additionally, a histogram normal probability plot of the residuals and the Shapiro-Wilk test were used to examine the data distribution, and the test indicated that the residuals were approximately normally distributed.

Linear regression was used to examine the association between HRQoL and predictor variables. The assumptions (normality test, correlation coefficient test, linearity test, outliers, multicollinearity, and homoscedasticity) of the statistical methods for all variables were tested. Variables such as hospitalization, triggering factors, and exercise were excluded from the analysis because they did not meet linearity. The variation around the regression line was tested by examining a plot of the standardized residuals versus standardized predicted values of the dependent variable, and it was constant for all values of xi for each X variable. The model's fitness was checked by the F-test of goodness of fit for linear regression.

The results of the regression analysis were expressed as an unstandardized coefficient, beta (β). Beta coefficients are measured in units of standard deviation and refer to the average change in the dependent variable for a unit increase in the predictor variable. The variables with a p-value of less than 0.2 in simple linear regression analysis were then entered into multi-variable linear regression to identify the HRQoL independent predictor variables. A p-value of < 0.05 at 95% confidence interval was used to declare statistical significance.

### Ethical considerations and consents for participation

Ethical clearance was obtained by the Institutional Review Board of the University of Gondar with a reference number of SOP/132/2021. A letter of permission was obtained from each hospital's clinical directorate. Study medical record numbers were used in place of patient names during the data collection and analysis process to conceal and safeguard participants' identities. All the materials for data collection were safely kept in a cabinet under lock and key. The principal investigator (PI) password-protected the databases to allow for only limited access. The data collected were used only for this study. Verbal and written consent were obtained after the purpose and objective of the study were explained to the selected participants. Moreover, all participants were informed that participation was on a voluntary basis, and they could withdraw from the study at any time if they were uncomfortable with the questionnaire. Any identifiers for the study participants were not recorded.

## Results

### Sociodemographic characteristics of the study population

The study included 409 participants (96.7% response rate) from 422 enrolled patients. The mean (±SD) age of the study participants was 49.82 (16.1) years, with the majority being between the ages of 35 and 64. Females made up more than half of them (60.1%). More than two-thirds of the respondents (71.9%) were married, and nearly three-quarters (73.3%) lived in cities. Most of the respondents (88.5%) used biomass fuel to cook their food, and charcoal and wood were the most commonly used food preparation materials (82.4%). The vast majority of participants in the study (87.5%) had never smoked (**Table 1**).

### Clinical characteristics and the triggering factors for asthma exacerbations

More than three-quarters (76%) of the study participants were diagnosed with asthma after the age of twelve. After the onset of the condition, the median (IQR) duration of the medications was 3 (1–6) years. The median (IQR) time to visit an emergency department after starting maintenance therapy was 11.95 (13.6) months. A higher number of patients had been taking the medication for 1–5 years. In the previous 12 months, nearly one-third (33.3%) of the patients were hospitalized, and 6.1% were admitted to the intensive care unit. In terms of medication, more than one-third (37.2%) used oral steroids. According to GINA-based severity classifications, 56.7% had moderately persistent asthma, and 26.2% had mildly persistent asthma (**Table 2**).

More than half (53.1%) had at least one episode of asthma exacerbation in the previous year, and 95.4% had at least one triggering factor for their exacerbations. More than half (53.5%) of them had exacerbated their symptoms during exercise, which is one of the leading causes of asthma exacerbation, followed by dust particles combined with cold weather (44.5%) and cold weather alone (22.5%). Only 8.1% of the female respondents had menstruation-induced asthma exacerbations (**Fig 2**).

**Table 1. Sociodemographic characteristics among patients with asthma attending ambulatory care units of selected referral public hospitals in Northwest Ethiopia, 2021 (N = 409).**

| Variables | | Frequency (%) | Mean (±SD) or Median (IQR) |
|---|---|---|---|
| Age (years) | 18–34 | 83 (20.3%) | 49.82 (±16.1) |
| | 35–64 | 238 (58.2%) | |
| | ≥ 65 | 88 (21.5%) | |
| Gender | Male | 163 (39.9%) | |
| | Female | 246 (60.1%) | |
| Residency | Urban | 300 (73.3%) | |
| | Rural | 109 (26.7%) | |
| Marital Status | Single | 41 (10%) | |
| | Married | 294 (71.9%) | |
| | Divorced | 16 (3.0%) | |
| | Widowed | 58 (14.2%) | |
| Education level | No formal education[a] | 174 (42.6%) | |
| | Primary education | 72 (17.6%) | |
| | Secondary education | 89 (21.8%) | |
| | Higher institute | 74 (18.1%) | |
| Occupation | Government employee | 115 (28.1%) | |
| | Farmer | 66 (16.1%) | |
| | Housewife | 112 (27.4%) | |
| | Merchant | 54 (13.2%) | |
| | Other[b] | 62 (15.1%) | |
| Average monthly income (Ethiopian birr) | - | - | 3450.6(±1015.7) |
| Source of healthcare cost coverage | Insurance | 173 (42.3%) | |
| | Free | 83 (20.3%) | |
| | Out of pocket | 153 (37.4%) | |
| Biomass use | Yes | 362 (88.5%) | |
| | No | 47 (11.5%) | |
| Fuel type used at home | Kerosene | 15 (3.7%) | |
| | Charcoal and wood | 337(82.4%) | |
| | Ethanol | 6 (1.5%) | |
| | Diesel-fuel | 13 (3.2%) | |
| Smoking status | Never smoker | 358 (87.5%) | |
| | Current smoker | 13 (3.2%) | |
| | Ex-smoker | 38 (9.3%) | |

[a]: Those unable to read and write; read and write due to informal education like religious teaching

[b]: Include individuals with daily labor, student, self-employed; SD; Standard Deviation, IQR; Inter Quartile Range

## Patterns of Anti-asthmatic medications

The drug usage pattern indicated that almost all patients (99%) used multiple-drug therapy (two or three drug combinations). More than three-fourths (76.5%) used Salbutamol puffs PRN and Beclomethasone puff Bid, followed by prednisolone (11.5%). The medication adherence rating scale revealed that 13.9% of the patients were highly adherent to the prescribed controller medication. At various stages, a large proportion of 305 (74.6%) respondents were on the optimal dosage of their asthma medication. For 71.1% of the subjects, asthma medication combination therapy was appropriate. Approximately 21.8% of prescribers did not follow the current guideline recommendations in their prescribing patterns. Almost three-fourths

**Table 2. Clinical characteristics among adult patients with asthma attending ambulatory care units of selected public referral hospitals in Northwest Ethiopia, 2021 (N = 409).**

| Variable | | Frequency (%) | Mean (±SD) or Median (IQR) |
|---|---|---|---|
| Age of onset (year) | < 12 years | 98 (24%) | |
| | ≥ 12 years | 311 (76%) | |
| Duration on medication (years median, (IQR)) | <1 year | 93 (22.7%) | 3(1–6) |
| | 1–5 years | 180 (44%) | |
| | 5–10 years | 87 (21.3% | |
| | >10 years | 49 (12%) | |
| Exacerbations in the last 12 months | Yes | 217 (53.1%) | |
| | No | 192 (46.9%) | |
| Average time to emergency visit after the start of maintenance therapy | - | - | 12.0 ±13.6 |
| Hospitalization in the last 12 months: | Yes | 136 (33.3%) | |
| | No | 273 (66.7%) | |
| Admitted to the ICU (intubated) in the last 12 months | Yes | 25 (6.1%) | |
| | No | 284 (93.9%) | |
| Oral steroid use | Yes | 152 (37.2%) | |
| | No | 257 (62.8%) | |
| Oral SABA (Salbutamol puff) use | Yes | 253 (61.9%) | |
| | No | 156 (38.1%) | |
| Asthma severity stage | Intermittent | 25 (6.1%) | |
| | Mild-persistent | 107 (26.2%) | |
| | Moderately persistent | 232 (56.7%) | |
| | Severely persistent | 45 (11%) | |

IQR; Inter Quartile Range, ICU; Intensive Care Unit, SABA, Short-acting beta agonist

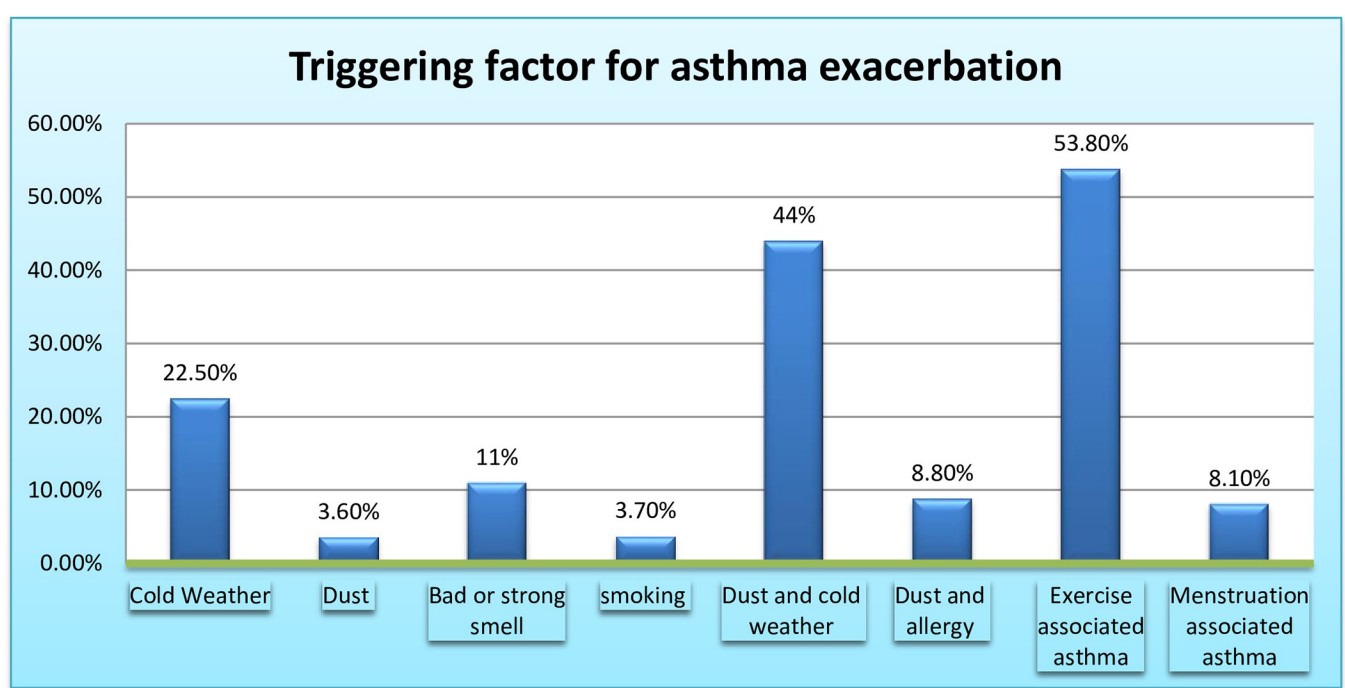

**Fig 2. Triggering factors for asthma exacerbations.**

**Table 3. The drug usage pattern among adult patients with asthma at ambulatory care units of selected public referral hospitals Northwest Ethiopia, 2021 (N = 409).**

| Variable | | Frequency (%) |
|---|---|---|
| Salbutamol puff PRN + Beclomethasone puff Bid | | 313 (76.5%) |
| Salbutamol puff PRN + Prednisolone oral daily | | 25 (6.1%) |
| Salbutamol puff PRN + Beclomethasone puff Bid + Prednisolone | | 47 (11.5%) |
| Salbutamol puff PRN + Beclomethasone puff Bid+ theophylline daily | | 6 (1.5%) |
| Salbutamol puff PRN+ Budesonide puff bid | | 4 (1%) |
| Theophylline +Salbutamol puff PRN | | 4 (1%) |
| Theophylline +Salbutamol puff PRN + prednisolone daily | | 1 (.2%) |
| Fluticasone puff +Salbutamol puff PRN | | 2 (0.5%) |
| Beclomethasone puff bid+ Salbutamol puff PRN+ Symicort | | 2 (.5%) |
| Almetamin | | 3 (.7%) |
| Oral SABA +salbutamol puff | | 2 (0.5%) |
| Medication adherent | High | 57 (13.9%) |
| | Low | 352 (86.1%) |
| Dose of the anti-asthmatic drug | Optimal | 305 (74.6%) |
| | Sub optimal | 104 (25.4%) |
| Appropriateness of drug selection based | Appropriate | 291 (71.1%) |
| on severity | Inappropriate | 118 (28.8%) |
| Healthcare providers adherent to | Yes | 320 (78.2%) |
| guideline | No | 89 (21.8%) |
| Patient information provided | Yes | 304 (74.3%) |
| | No | 105 (25.7%) |
| Patient relationship with healthcare provider | Good | 375 (91.7%) |
| | Poor | 34 (8.3%) |

PRN; Take as needed, bid; Two-times daily, SABA; Short acting beta agonist

(74.3%) of those polled felt they had received adequate education about asthma and its management. Participants had a good relationship with healthcare providers and were satisfied with their care in 91.7% and 89.5% of cases, respectively (**Table 3**).

## Co-morbidities and concurrent medications

According to the Charlson Co-morbidity Index (CCI), nearly sixty percent of the respondents were classified as mild, and approximately forty percent of the individuals had co-morbidities. Concurrent medications were prescribed to more than one-third of the patients (38.3%) (**Table 4**).

In this study, many co-morbidities were documented. A large proportion of participants (21.5%) had cardiovascular disease, followed by diabetes mellitus (10%), and cardiovascular drugs were the two most commonly used classes of concurrent medications. Endocrine drugs accounted for 18.6%, with endocrine drugs accounting for 10.5% (**Fig 3**).

## Beliefs about medicines in the study population

The mean (SD) belief score for anti-asthmatic medications measured using the Specific-Necessity Scale and the Specific Concerns Scale out of 25 patients was 17.9 (4.4) and 16.5 (4.9), respectively. The participants' overall mean (SD) score was 3.46 (0.54) out of five (**Table 5**).

**Table 4. Charlson co-morbidity index, co-morbidity history and concurrent medication use among patients with asthma.**

| Variables | | Frequency (%) |
|---|---|---|
| CCI | Mild | 241 (58.9%) |
| | Moderate | 132 (32.3%) |
| | Severe | 36 (8.8%) |
| Co-morbidities | Yes | 162 (39.6%) |
| | No | 244 (59.4%) |
| Concurrent medication use | Yes | 157 (38.4%) |
| | No | 252 (61.6%) |

CCI; Charlson comorbidity index

### Health-related quality of life outcomes

The mean Mini-AQLQ score was 4.1 (±0.9). Emotional, environmental, and symptom stimulation were the individual domains with the lowest mean score. Out of the 409 participants, 242 (59.2%) had good HRQoL with mean scores of 4.1 or higher. Using linear regression, additional data analysis was performed to identify the independent predictor of HRQoL. Pearson correlation analysis was used to assess the relationship between participants' min-AQLQ and ACT scores, and the results revealed that our dependent variables were strongly associated (r = 0.59; p = 0.01). The Mini-AQLQ score increased as the level of asthma control increased (**Table 6**).

### Factors associated with health-related quality of life

A linear regression analysis was used to identify potential variables influencing the health-related quality of life of asthma patients. The fitness of the linear regression model was tested and found to be significantly associated (F = 11.68; p = 0.001). A multivariable linear

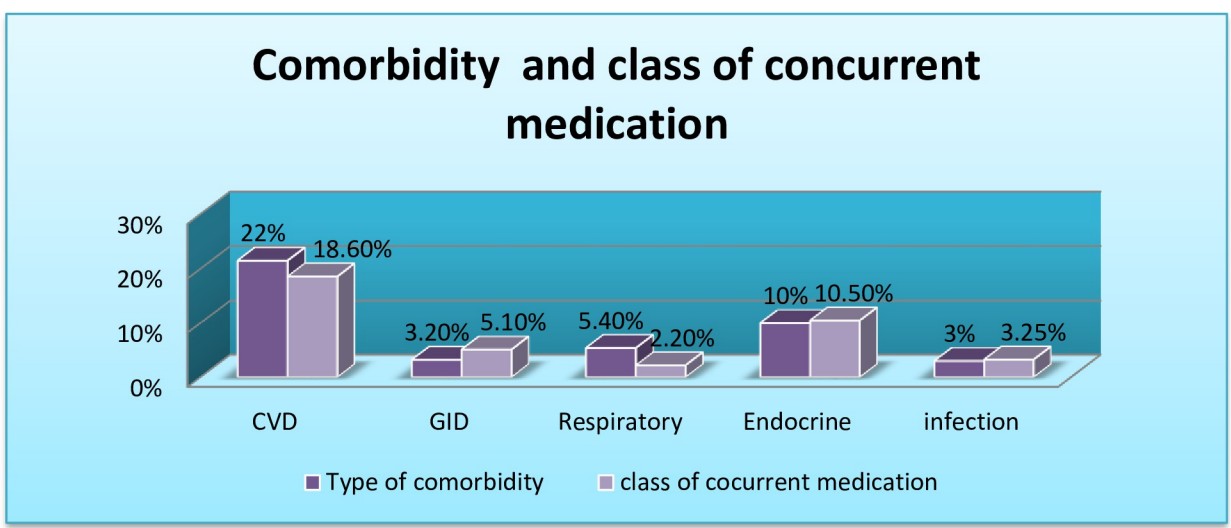

CVD: cardiovascular     GI: gastrointestinal

**Fig 3. The type of comorbidity and the commonly used class of concurrent medication among adult asthmatic patients.**

**Table 5. The percentage of respondents agreed/strongly agreed with their medication beliefs on their people who have asthma medications.**

| Items in the two scales | | Frequency (%) | Mean (±SD) score of each item | Overall mean score for specific scales |
|---|---|---|---|---|
| Specific-Necessity Scale | My health at present depends on my asthma medicines | 351 (85.8%) | 3.9 (±0.7) | 17.9±4.4 |
| | My life would be impossible without my asthma medication | 175 (42.8%) | 3.2 (±1.1) | |
| | Without my asthma medication I would be very ill | 273 (66.7%) | 3.6 (±0.9) | |
| | My health in the future will depend on my asthma medication | 250 (61.1%) | 3.5 (±0.9) | |
| | My asthma medication protects me from becoming worse | 323 (79%) | 3.7 (±0.8) | |
| Specific-Concerns Scale | Having to take medication worry me | 154 (38%) | 3.4 (±1.0) | 16.5±4.9 |
| | I sometimes worry about the long-term effects of my asthma medication | 244 (59.7%) | 3.4 (±1.0) | |
| | My asthma medication is a mystery to me | 223 (53.5%) | 3.6 (±1.0) | |
| | My asthma medication disrupts my life | 253 (61.8%) | 3.6 (±1.0) | |
| | I sometimes worry about becoming too dependent on my asthma medication | 315 (77%) | 3.6 (±0.9) | |
| **Over all medication belief mean score** | | - | **3.54± (0.5)** | **17.2(±4.7)** |

SD; Standard Deviation

regression analysis identified occupation, healthcare service, the role of patient enablement, adherence to guidelines, total asthma control score, and belief in anti-asthmatic medication as factors potentially associated with HRQoL. According to the regression results, the model explained 45.6% of the variance, and the variance inflated factor for all variables was less than five.

Being a housewife had a lower HRQoL than being a government employee, with a B score of 0.21 times ($\beta$ = -0.21, 95% CI (-0.39, -0.01) indicating that on average, housewife asthmatic patients had 0.21 times lower HRQoL than government employers; using insurance for health care services ($\beta$ = 0.15, 95% CI (0.010, 0.29), p = 0.036) indicating that on average, insurance users have 0.148 times higher HRQOL than those who paid for services, with a B score of 0.48. For every increase in asthma control score, the patient's HRQOL increases by 0.14 times (95% CI (0.09, 0.17), p < 0.001) percentage point; on average, patients whose health care provider does not adhere to guidelines have 0.30 times (95% CI (-0.47, -0.15), p < 0.001) lower HRQOL than patients whose care providers adhere to guidelines; and for every increase in belief in an anti-asthmatic medication score, the HRQOL decreases by 0.23 times (95% CI (-0.36, -0.101) percentage point. The variables listed above were found to be significant predictors of health-

**Table 6. The mini- asthma QoL outcome measures.**

| Variable | Mean (±SD) score | |
|---|---|---|
| Symptoms domain | 3.9 (±1.0) | |
| Environmental domain | 3.85(±1.0) | |
| Emotional domain | 3.8 (±1.0) | |
| Activity domain | 4.7 (±1.0) | |
| **Overall Mini-AQLQ mean (±SD) score out of 5** | **4.1(±0.9)** | |
| Overall level of quality of life | | |
| Good | 242 (59.2%), 95% CI (54.1,64.1) | |
| Poor | 167 (40.8%), 95% CI (35.9,45.2) | |

Mini-AQLQ, Mini-Asthma Quality of Life Questionnaire; SD; Standard Deviations

**Table 7. Simple and multiple linear regression analysis for determining of health-related quality of life among patients with asthma in selected hospitals in North Northwest Ethiopia (N = 409).**

| Variable | | SLR β (95% CI) | p-value | AdjR²% | MLR β (95% CI) | p-value |
|---|---|---|---|---|---|---|
| Total asthma control levels | | 0.12 (0.11,0.14) | **<0.001** | 34.6 | 0.14(0.09, 0.17) | **0.001**\*\* |
| Medication burden | | -0.08(-0.16,0.01) | **0.068** | 0.6 | -0.03(-0.12, 0.06) | 0.493 |
| CCI | | -0.05(-0.10,0.01) | **0.139** | 0.3 | 0.01(-0.05,0.06) | 0.664 |
| Beliefs in medication Mean score | | -0.393(-0.54, -0.24) | **<0.001** | 6 | -0.23-(0.358,0.101) | **0.001**\* |
| Sex | Female | -0.237(-0.40, -.07) | **0.006** | 1.6 | -0.04(0.20,0.12) | 0.613 |
| | Male ^R | 0 | | | 0 | |
| Marital status | Single | 0.24 (-0.04,0.52) | **0.09** | 0.7 | 0.14(-0.09,0.36) | 0.230 |
| | Divorced | -0.11 (-0.54,0.32) | 0.613 | | -0.19(-0.58,0.20) | 0.341 |
| | Window | -0.13 (-0.36,0.12) | 0.324 | | 0.13(-0.16,0.41) | 0.383 |
| | married ^R | 0 | | | 0 | |
| Level of education | Informal | 0.05 (-0.13,0.24) | 0.570 | 0.7 | -0.068(-0.27,0 13) | 0.511 |
| | High level | 0.27(0.03,0.5) | **0.202** | | 0.082(-0.22,0.38) | 0.603 |
| | Low-level ^R | 0 | | | 0 | |
| Occupation | Farmer | 0.06(-0.18,0.30) | 0.613 | 6.3 | -0.01(-0.22,0.22) | 0.960 |
| | Private work | -0.33(-0.60,0.05) | **0.022** | | -0.17(-0.35,0.02) | 0.080 |
| | House wife | -0.56(-0.82,0.31) | **0.001** | | -0.21(-0.39, -0.012) | **0.037**\* |
| | Gov,t employ ^R | 0 | | | 0 | |
| Biomass use | No | 0.24 (-0.03,0.50) | **0.080** | 0.8 | 0.07 (-0.41, 0.54) | 0.781 |
| | Yes ^R | 0 | | | 0 | |
| Health care Service | free | 0.10 (-0.13,0.33) | 0.383 | 1.5 | -0.22 (-0.41,0.04) | 0.212 |
| | Insurance | 0.114 (0.42, 0.491 | **0.020** | | 0.148 (0.01, 0.29) | **0.036**\* |
| | Payment ^R | 0 | | | 0 | |
| Year of onset | < 12 years | -0.21(-0.49,0.10) | **0.003** | 2 | -0.08 (-0.23, 0.08) | 0.321 |
| | ≥ 12year ^R | 0 | | | 0 | |
| Role of enablement | high | 0.79 (0.62,0.95) | **<0.001** | 17.4 | 0.39 (0.25, 0.54) | **0.001**\*\* |
| | low ^R | 0 | | | 0 | |
| Patient satisfaction by service | Yes | 0.31 (0.12,0.66) | **0.005** | 1.7 | 0.05 (-0.11,0.29) | 0.704 |
| | No ^R | 0 | | | 0 | |
| Relationship to HCP | Good | 0.78 (0.491,1.077) | **<0.001** | 6.1 | 0.01 (-0.19,0 38) | 0.523 |
| | Poor ^R | 0 | | | 0 | |
| Adherence | Adherent | 0.23 (0.13,0.33) | **<0.001** | 4.4 | 0.02 (-0.13,0.08) | 0.64 |
| | Non-adherent ^R | 0 | | | 0 | |
| Comorbidity | Yes | 0.79 (-0.06,0.21) | 0.262 | 0.1 | -0.050(-0.8, 0.078) | 0.443 |
| | No ^R | 0 | | | 0 | |
| HCP adherence to guideline | No | -0.58 (-)(0.79,0.38) | **<0.001** | 7.8 | -0.30(-0.47,-0.14) | **0.001**\*\* |
| | Yes ^R | | | | 0 | |

SLR, simple linear regression; MLR, multiple linear regressions; ^R; Reference; \*, significant $p < 0.05$\*\* $p < 0.001$ Confidence interval, adjusted $R^2 = 45.7$, F = 11.68; $P < 0.001$, VIF<5, CCI, Charlson Comorbidity Index

related quality of life, but the other variables lost their effect after multiple variable analysis (**Table 7**).

## Discussion

This multicenter study was the initial study to assess HRQoL and determinants of HRQoL in patients with asthma in the study settings. As a result, we hope it will be helpful for the

healthcare providers to tailor the management approaches for these patients, and it will be a source of data for future research in the area.

This study reported that the overall mean (±SD) HRQoL was 4.1 ±0.9 (out of 7). The HRQoL status of the current study revealed that about 60% of the patients with asthma had a mean (±SD) of 4.1 ±0.9 or more than this score. This finding agrees with the study results reported in Iran, in which 53% of patients had a good quality of life [36]. In contrast to this finding, it was much higher than the findings from Pakistan, where only 28.6% of the study subjects had a good quality of life [37]. The possible explanation for the higher number of patients who had a good quality of life in this study might be the mean age levels of the respondents, whose ages were in the middle ranges compared with those in Pakistan, where their age ranges were found in older groups.

For the factors associated with HRQoL, the following variables were disclosed: health-insured patients, an advanced role of patient enablement, and improved asthma control levels that significantly increased HRQoL status. However, as being homemakers, patients who were not treated as per the guidelines and individuals who positively believed in their medication significantly declined in HRQOL status.

According to the results of this study, patients who used health insurance have significantly better HRQoL. This positive association between health insurance and HRQoL was supported by other studies [12, 24, 25, 38–43]. The possible explanation for this positive association might be mainly due to high levels of adherence, reduced out-of-pocket costs of drugs, and increased use of treatment as per guidelines among health insurance participants. Additionally, it might be due to insurance users having a lower risk of asthma exacerbation because this reduces stress, increases adherence to medication, and would impact hospital cost minimization [44].

This study revealed that the role of patient enablement was significantly associated with Mini-AQoL, which is supported by some evidence that the mPEI may be sensitive enough to detect changes in the patients' health-related quality of life, as shown by the United Kingdom study [33]. There are several possible reasons why measures of patient enablement are significantly associated with the quality of life. Although self-management training programs may bring about only mild-to-moderate outcomes for selected chronic diseases [45], they may improve asthma control in patients compared with routine care [46]. Similarly, they trust their treatment and improving adherence to therapeutic plans [47], which indirectly leads to increased health-related quality of life in patients with asthma.

A high level of asthma control was identified as a positive independent predictor of HRQoL. These findings correspond with those of multiple relevant studies conducted worldwide; for instance, in Brazil and the United Kingdom, researchers found that the degree to which asthma was controlled had a significant impact on a patient's HRQoL [15, 48, 49]. Asthma control reflects the disease's effect on a patient as reflected by fluctuations in their symptoms, limitations in their range of activities, and their environmental and emotional functioning. The link between asthma control and HRQoL was also highlighted by the findings of an Italian study, which discovered that nearly one-third of their population had optimal HRQoL, which was not associated with the duration of severity of asthma or rhinitis but with the degree of asthma control [38]. Poor asthma control was the only factor that independently impacted HRQoL in a similar study conducted in France and the United Kingdom [41]. These consistent outcomes confirm that asthma control is indeed the single most important determinant factor of an individual's HRQoL. The negative impact of asthma on patients' HRQoL could be reduced if patient care focused on achieving good control of the disease. To achieve optimal asthma control, variables that significantly affect populations need to be identified and addressed so that patients can lead near-normal lives.

The findings of this study indicate that individuals with a history of occupational risk exposure to asthma triggers, like housewives' workers, had poor HRQoL that was associated with asthma. This finding interrelates with a similar study in the United States, which indicated that individuals with work-related asthma were significantly more likely to have poor HRQoL compared with those without work-related asthma [50]. The possible explanation for this is that being a housewife increases exposure to asthma triggers such as baking, rubber or plastic work, cleaning, spray painting, and food processing. Additionally, being a female makes you more exposed to asthma triggers due to the natural hormone associated with estrogen and increased asthma exacerbations during menstruation [51], which leads to a negative impact on HRQOL.

This study found a significant association between HRQoL and guideline usage by the prescribers. Our results were comparable to those of a study based on GINA guidelines, where investigators found that well-controlled patients who had achieved guideline-based asthma control reported consistently higher overall HRQoL than their uncontrolled counterparts, in whom guidelines were not followed [13]. In another cross-sectional study, where half the treatment regimens were considered non-adherent to guideline recommendations, only those patients whose treatment was in accordance with guidelines had significantly higher HRQoL [16]. In this study, 21% of the patients had not been treated based on guidelines; those whose guidelines were not followed either had their controller medications prescribed once daily, were on SABA only instead of combination with controller medication, or their asthma severity level required a step-up to a higher dose or the addition of other medications to those they had been put on. Therefore, non-adherence to guidelines would reduce the level of asthma control and, by extension, the HRQoL of patients with asthma.

In this study, medication belief in patients with asthma was identified as an independent predictor of HRQoL. Patients with high medication belief had poor HRQoL, according to the findings. The link could be explained by the fact that, in this study, a significant number of patients were more concerned about the adverse effects and negative consequences of their medications based on their responses to MBQ. Most of them indicated a higher level of concern on the MBQs' specific-concern scales, with a comparable score of specific-concern and specific-necessity. As a result, patients with a high level of concern regarding negative effects might have poor medication adherence and poor HRQoL associated with it. As a result, healthcare providers are recommended to be highly involved in counsel regarding the benefits and harms of the medications. However, this finding is in contradiction with a finding from another study [52], which showed that higher medication beliefs resulted in high medication adherence and quality of life. This discrepancy might be because of the differences in patients' sociodemographic, perceptions, and knowledge of the advantages, adverse effects, and/or harmful effects of the medications.

## Clinical implications of the study

Assessment of HRQoL in patients with asthma is pertinent to clinical practice because treatment planning and progression are focused on the patient rather than the disease. Although HRQoL in patients with asthma improved when the disease was controlled, optimal scores were not always observed. The achievement of asthma control does not necessarily show the achievement of maximal HRQoL. Therefore, this study will be an important input to assist an existing effort to improve HRQoL and target the associated factors. Additionally, the findings will provide good evidence in the study setting to plan interventional strategies, a body of knowledge for further study that might be conducted on a related topic, or for organizations working with asthma patients, and might have important clues to characterize and stratify

patients at follow-up care and optimize care based on pertinent precipitants. Through longitudinal research, future research should focus on investigating the predictors of HRQoL among patients with asthma and testing different interventions or strategies for overcoming the determinants of poor HRQoL.

## Strengths and limitations of this study

To the best of the authors' knowledge, this is the first study to explore the determinants of health-related quality of life among patients with asthma in Northwest Ethiopia. A systematic random sampling technique was used. This could have reduced the source of bias in the study. However, some responses in the questionnaires were patient-subjective reports, which could have resulted in social desirability bias either by under-reporting or exaggerating. The study had a cross-sectional design that could make it difficult to identify whether the cause or effect happened first. Researchers would also recommend to investigating factors of HRQoL in patients with asthma in resource-limited settings using larger population samples on prospective designs.

## Conclusion

Our study revealed that slightly over half of the study participants showed good HRQoL. Health-related quality of life outcomes among patients were largely dependent upon the degree to which asthma was controlled. Being a housewife, having a non-adherent healthcare provider, and high MBQ score significantly decreased HRQoL. However, insurance users for healthcare services, a higher patient enablement and good asthma control score significantly increased HRQoL. As a result, patient education on patients' behavior, medication adherence, and asthma control would be expected from healthcare providers.

## Supporting information

**S1 File. The data set used to analyze and generate data of the manuscript.**
(SAV)

## Acknowledgments

We forward our appreciation to the clinical directors of all hospitals for their positive cooperation to conduct this research. Our special appreciation goes to the data collectors and study participants for their volunteer participation.

## Author Contributions

**Conceptualization:** Eyayaw Ashete Belachew.

**Data curation:** Ashenafi Kibret Sendekie, Sumeya Tadess, Mekuriaw Alemayehu.

**Formal analysis:** Eyayaw Ashete Belachew, Ashenafi Kibret Sendekie, Sumeya Tadess, Mekuriaw Alemayehu.

**Funding acquisition:** Eyayaw Ashete Belachew.

**Investigation:** Ashenafi Kibret Sendekie.

**Methodology:** Eyayaw Ashete Belachew, Ashenafi Kibret Sendekie, Sumeya Tadess, Mekuriaw Alemayehu.

**Resources:** Sumeya Tadess.

**Software:** Eyayaw Ashete Belachew, Ashenafi Kibret Sendekie, Sumeya Tadess, Mekuriaw Alemayehu.

**Supervision:** Mekuriaw Alemayehu.

**Validation:** Eyayaw Ashete Belachew, Ashenafi Kibret Sendekie, Sumeya Tadess, Mekuriaw Alemayehu.

**Visualization:** Ashenafi Kibret Sendekie, Sumeya Tadess.

**Writing – original draft:** Eyayaw Ashete Belachew, Ashenafi Kibret Sendekie, Sumeya Tadess, Mekuriaw Alemayehu.

**Writing – review & editing:** Eyayaw Ashete Belachew, Ashenafi Kibret Sendekie, Sumeya Tadess, Mekuriaw Alemayehu.

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
