## [Decision Letter · Decision Letter 0]

8 Jan 2023

PONE-D-22-30759Health-related quality of life and its associated factors among patients with asthma: A multi-centered cross-sectional study in selected referral hospitals in Northwest EthiopiaPLOS ONE

Dear Dr. Belachew,

Thank you for submitting your manuscript to PLOS ONE. After careful consideration, we feel that it has merit but does not fully meet PLOS ONE’s publication criteria as it currently stands. Therefore, we invite you to submit a revised version of the manuscript that addresses the points raised during the review process.

We look forward to receiving your revised manuscript.

Kind regards,

Yen-Ming Huang, PhD

Academic Editor

PLOS ONE

Journal Requirements:

 "The study was not funded". 

Reviewers' comments:

Reviewer's Responses to Questions

**Comments to the Author**

1. Is the manuscript technically sound, and do the data support the conclusions?

Reviewer #1: Yes

Reviewer #2: Yes

2. Has the statistical analysis been performed appropriately and rigorously? 

Reviewer #1: Yes

Reviewer #2: No

3. Have the authors made all data underlying the findings in their manuscript fully available?

Reviewer #1: Yes

Reviewer #2: Yes

4. Is the manuscript presented in an intelligible fashion and written in standard English?

Reviewer #1: Yes

Reviewer #2: No

5. Review Comments to the Author

Reviewer #1: 1. Authors state that they conducted a cross-sectional study among patients who were followed up at selected hospital clinics (Page 10 of 37, line 28). There is no need of including critically ill and inpatients as part of exclusion criteria

2. Page 11 of 37, line 11: Authors state that "Respondents were assessed in proportion to the number of patients admitted in the respective hospitals". This is not clear. Authors should add details as to why "admitted patients" were included in the narrative

3. The study design was cross-sectional . This study design does not usually involve power calculations. Authors have stated in the strengths section that "the study was well powered". This is not right, given the fact that multiple study tools were used (the fact not considered in sample size calculation); sample size appears relatively small

4. For methods sections. For Beliefs about asthma medication questionnaire (BMQ) and modified patient enablement index (mEPEI). Authors should provide more details on these tools. What is the total score?. Which is best for BMQ scale, high or low belief score?

5. The Cronbach alpha score for BMQ was 0.73, translating as acceptable, unlike other tools scores that were either Vey good or excellent in internal consistency. Could the alpha score of BMQ be the reason for the finding that belief about medication score was negatively affecting HRQOL?

6. There is a discrepancy on interpretation of association between BMQ score and HRQL in the results to that provided in the discussion. Authors should correct this discrepancy

Reviewer #2: The objective of this study was to evaluate the health-related quality of life (HRQoL) and its associated factors among adult patients living with asthma who had been attending selected public referral hospitals in Northwest Ethiopia. A multicenter facility-based cross-sectional study was conducted in selected hospitals in Northwest Ethiopia. Participants were enrolled into the study using a systematic random sampling technique. HRQoL was measured by using the asthma-specific quality of life tool Mini-Asthma Quality of Life Questionnaire (Mini-AQLQ). A total of 409 patients were included in the final analysis. Regarding determinants of HRQoL, asthma control score, insurance user, high role of patient enablement, belief in asthma medication, health care provider non-adherence to guidelines, and being house wife were the significant predictor of HRQoL. They concluded that HRQoL among adults with asthma was largely dependent on the level of asthma control.

This is very interesting and important topic. Authors used appropriate methodological approach to achieve the study objectives. However, certain items need to be clarified.

Response rate was unusually high, 96.7%. How authors explain this finding?

Why authors used p less or equal 0.2 as criterion for inclusion in multivariate models? P value of 0.05 is usual. Therefore, I suggest authors to perform multivariate analysis again, by using p of 0.05 as criterion for selection of variables which were significant in univariate analysis.

The tables are not clear enough and should be corrected.

English should be corrected by a native speaker.

6. PLOS authors have the option to publish the peer review history of their article (what does this mean?). If published, this will include your full peer review and any attached files.

Reviewer #1: **Yes: **Godfrey Mutashambara Rwegerera

Reviewer #2: No

---

## [Author Response · Author response to Decision Letter 0]

11 Jan 2023

Responses to the review’s comments

Dear PLOS ONE editor,

Thank you for giving us the opportunity to submit a revised draft of the manuscript, and we would also like to thank you for your crucial comments on our paper (PONE-D-22-30759). We are very concerned and have combined all the suggested comments provided, which we believe strengthen our paper, and we hope this will render our paper eligible for consideration for publication in your reputed journal. We appreciate the time and effort that you and the reviewers dedicated to providing feedback on our manuscript and are grateful for the insightful comments and valuable improvements to our paper.

The authors would like to inform you that we have addressed the comments and recommendations made by both reviewers and the editor, point by point. In addition, throughout our revision, we made our best corrections too. All changes in the revised manuscript are highlighted using tracking changes within the manuscript. Please see below, in blue, for a point-by-point response to the reviewers’ comments and concerns. All page numbers refer to the revised manuscript file with tracked changes.

Comments from the editor:

#1---- Journal Requirements:

Author response: Thank you for your suggestions to ensure Plos One’s reequipments. Considering your valuable recommendation, we have verified and revised that our submission ensured the journal’s requirements. 

#2...... Thank you for stating the following financial disclosure: 

 "The study was not funded". 

Author response: Thank you for your suggestions to clarify the financial disclosure. As we mentioned earlier, we ensured that we did not access any funding, either financial or material, from any organization. Therefore, based on Plos One’s revision, we have revised the statement as "The authors received no specific funding for this work." We have also included the revised statement in our cover letter.

#3.... Please amend your list of authors on the manuscript to ensure that each author is linked to an affiliation. Authors’ affiliations should reflect the institution where the work was done (if authors moved subsequently, you can also list the new affiliation stating “current affiliation:.” as necessary).

Author response: Thank you for your important comments, ensuring the affiliation of the second author, which was missed. The affiliation was the same with the first and third authors, and we revised it accordingly.

#4...... Please include captions for your Supporting Information files at the end of your manuscript, and update any in-text citations to match accordingly.

Author response: Thank you for making the important suggestion to include a caption for the supporting information. Thus, we have included it at the end of our manuscript and cited it with appropriate text in the data availability statement section accordingly.

Response to Reviewers’ comments

Reviewer 1:

#1…... Authors state that they conducted a cross-sectional study among patients who were followed up at selected hospital clinics (Page 10 of 37, line 28). There is no need of including critically ill and inpatients as part of exclusion criteria.

Author response: We, the authors, are very grateful for your constrictive and impactful comments in general, which we believe can improve the quality of the paper.

Regarding the inclusion and exclusion criteria, our intention was to disclose that those patients who were initially on follow-up and then transferred to inpatient status during the data collection period were excluded. Based on your valuable comments and the fact that inpatients were not part of our study population from the outset, we revised and corrected it accordingly (see page 4, line 25-27).

#2…... Page 11 of 37, line 11: Authors state that "Respondents were assessed in proportion to the number of patients admitted in the respective hospitals. This is not clear. Authors should add details as to why "admitted patients" were included in the narrative.

Author response: Thank you for your comments and concerns. It was to show how many participants were included from each hospital because they were recruited from a multicenter (three hospitals) study. However, admitted individuals were not included, and the number of participants from each hospital was allocated proportionally to the number of patients admitted or having follow-up at each hospital in the previous three months. So, we have revised and made it clear accordingly (see page 5, line 1-14).

#3…. The study design was cross-sectional. This study design does not usually involve power calculations. Authors have stated in the strengths section that "the study was well powered". This is not right, given the fact that multiple study tools were used (the fact not considered in sample size calculation); sample size appears relatively small

Author response: Thank you for your comments and concerns. Of course, we used a simple population proportion formula to calculate the sample size, but it was just to inform relatively comprehensive findings using validated tools compared with other studies. Therefore, we have revised and corrected it according to your comments (see the discussion ).

#4…. For methods sections. For Beliefs about asthma medication questionnaire (BMQ) and modified patient enablement index (mEPEI). Authors should provide more details on these tools. What is the total score? Which is best for BMQ scale, high or low belief score?

Author response: Thank you for your valuable comments to clarify the tools mentioned. Based on your recommendation, we have revised and incorporated their expected minimum and maximum total score. We have also shown the relationships between the scores and the outcomes on belief and patient enablement (see page 6, line 16-27).

#5…. The Cronbach alpha score for BMQ was 0.73, translating as acceptable, unlike other tools scores that were either Very good or excellent in internal consistency. Could the alpha score of BMQ be the reason for the finding that belief about medication score was negatively affecting HRQOL?

Author response: We certainly appreciate your request to be clear on this point. Even though the alpha score of BMQ is lower than the other tools used in our study still it is acceptable. The negative association of BMQ and HRQoL in our study is as we see it in the discussion section a significant number of patients were more concerned about the adverse effects and negative consequences of their medications based on their responses to MBQ. The majority of our participants had indicated a higher level of concern on the MBQs' specific-concern scales. As a result, patients with a high level of concern regarding negative effects might have poor medication adherence and poor HRQoL associated with it.

Therefore, we authors have believed that the alpha score of BMQ is not the reason for the negative association with HRQOL. 

#6…... There is a discrepancy on interpretation of association between BMQ score and HRQL in the results to that provided in the discussion. Authors should correct this discrepancy

Author response: Thank you very much for these important corrections. We have revised and corrected the discrepancies related to interpretation of the association of BMQ and HRQoL (see page 22, line 4-17). 

Reviewer 2 

The objective of this study was to evaluate the health-related quality of life (HRQoL) and its associated factors among adult patients living with asthma who had been attending selected public referral hospitals in Northwest Ethiopia. A multicenter facility-based cross-sectional study was conducted in selected hospitals in Northwest Ethiopia. Participants were enrolled into the study using a systematic random sampling technique. HRQoL was measured by using the asthma-specific quality of life tool Mini-Asthma Quality of Life Questionnaire (Mini-AQLQ). A total of 409 patients were included in the final analysis. Regarding determinants of HRQoL, asthma control score, insurance user, high role of patient enablement, belief in asthma medication, health care provider non-adherence to guidelines, and being house wife were the significant predictor of HRQoL. They concluded that HRQoL among adults with asthma was largely dependent on the level of asthma control.

This is very interesting and important topic. Authors used appropriate methodological approach to achieve the study objectives. However, certain items need to be clarified.

#1…. Response rate was unusually high, 96.7%. How authors explain this finding?

Author response: Thank you for your comments and concerns regarding the response rate. We understand your concern because this figure was high. As we explained in the method section the data was collected by questionnaire-based face-to-face interviews and before we began collecting data, we informed the eligible individuals about the purpose of the study and the implications of their participation in the study. We informed them that their participation in this study would cause no harm and could even help them overcome a barrier to their quality of life based on the findings. At the same time, we disclosed that because of their participation, nothing was rewarded to them. And it was volunteer-based involvement. Furthermore, patients with chronic illness (asthma) may have more knowledge and experience with the ethical aspects of scientific research and may be more willing to participate. 

#2…Why authors used p less or equal 0.2 as criterion for inclusion in multivariate models? P value of 0.05 is usual. Therefore, I suggest authors to perform multivariate analysis again, by using p of 0.05 as criterion for selection of variables which were significant in univariate analysis.

Author response: Thank you for your suggestions and concerns regarding a cutoff point to include variables in the multivariable models. In fact, we shared your concern, and evidence may suggest a different cutoff point. We authors are grateful for your deep insight and let us give a clear justification for this issue. 

# Generally, stepwise variable selection, univariable screening, and any method that eliminates “insignificant” predictor variables from the final model cause a multitude of serious problems related to bias, significance, improper confidence intervals, and multiple comparisons. Stepwise variable selection should be avoided unless backward elimination is used with an alpha level of 0.5 or greater. 

#On the other hand, the univariable screening producers, and statisticians have stated that several literatures could provide pragmatic recommendations for the actual statistical approaches on the application of univariable selection procedures. Usually, the selection trends on the univariate screening for determining further multivariate model inclusion are found poor method. Nevertheless, several researchers recommend the probability (p-value<0.20) cutoff points for screening variables in the univariate model to build the multivariate model. 

#However, the common pragmatic statistical trends also used the assumptions of regression model that the variables with the probability (P-value less than 0.20) are used to select out univariable in the primary approaches. Therefore, if a predictor variable has a probability value (P-value <0.20), we consider it for a further multivariate model that for adjusted rations. On the other hand, the probability (P-value <0.05) is the cut-off point for the test of significance. 

We used a p-value of < 0.2 not to declare a statistically significant variable but rather to include potential variables, which may have a significant association with HRQoL, for further analysis in the multivariable model to appreciate the strength of association with HRQoL by taking other variables into account. As a result, variables that fulfilled the assumptions of the model and had a p value of < 0.2 in the univariable model were included in the multivariable model and further analyzed.

Finally, we used a p value of < 0.05 to declare a significant association between independent variables and HRQoL. If you are not comfortable with our justification and if you have any other recommendations, we are happy to do so.

3#... The tables are not clear enough and should be corrected.

Author response: Thank you for your comments to clear the tables. We have found some ambiguous points on the table, and based on your important comments, we have made the tables clear (see tables).

4#...... English should be corrected by a native speaker.

Author response: Thank you very much for your important suggestions and comments. Based on your suggestions, we have revised the whole manuscript and made our best changes accordingly. We hope you have found it improved now.

---

## [Decision Letter · Decision Letter 1]

31 Jan 2023

Health-related quality of life and its associated factors among patients with asthma: A multi-centered cross-sectional study in selected referral hospitals in Northwest Ethiopia

PONE-D-22-30759R1

Dear Dr. Belachew,

We’re pleased to inform you that your manuscript has been judged scientifically suitable for publication and will be formally accepted for publication once it meets all outstanding technical requirements.

Kind regards,

Yen-Ming Huang, PhD

Academic Editor

PLOS ONE

Additional Editor Comments (optional):

Reviewers' comments:

Reviewer's Responses to Questions

**Comments to the Author**

1. If the authors have adequately addressed your comments raised in a previous round of review and you feel that this manuscript is now acceptable for publication, you may indicate that here to bypass the “Comments to the Author” section, enter your conflict of interest statement in the “Confidential to Editor” section, and submit your "Accept" recommendation.

Reviewer #1: All comments have been addressed

Reviewer #2: All comments have been addressed

2. Is the manuscript technically sound, and do the data support the conclusions?

Reviewer #1: Yes

Reviewer #2: Yes

3. Has the statistical analysis been performed appropriately and rigorously? 

Reviewer #1: Yes

Reviewer #2: Yes

4. Have the authors made all data underlying the findings in their manuscript fully available?

Reviewer #1: Yes

Reviewer #2: Yes

5. Is the manuscript presented in an intelligible fashion and written in standard English?

Reviewer #1: Yes

Reviewer #2: Yes

6. Review Comments to the Author

Reviewer #1: (No Response)

Reviewer #2: Authors answered all my questions with appropriate explanations. I have no any other concerns and additional questions.

7. PLOS authors have the option to publish the peer review history of their article (what does this mean?). If published, this will include your full peer review and any attached files.

Reviewer #1: **Yes: **Godfrey Mutashambara Rwegerera

Reviewer #2: **Yes: **Tatjana Pekmezovic

---

## [Editor Report · Acceptance letter]

5 Feb 2023

PONE-D-22-30759R1 

Health-related quality of life and its associated factors among patients with asthma: A multi-centered cross-sectional study in selected referral hospitals in Northwest Ethiopia 

Dear Dr. Belachew:

I'm pleased to inform you that your manuscript has been deemed suitable for publication in PLOS ONE. Congratulations! Your manuscript is now with our production department. 

Kind regards, 

on behalf of

Dr. Yen-Ming Huang 

Academic Editor

PLOS ONE